# Prior Routine Use of Non-Steroidal Anti-Inflammatory Drugs (NSAIDs) and Important Outcomes in Hospitalised Patients with COVID-19

**DOI:** 10.3390/jcm9082586

**Published:** 2020-08-10

**Authors:** Eilidh Bruce, Fenella Barlow-Pay, Roxanna Short, Arturo Vilches-Moraga, Angeline Price, Aine McGovern, Philip Braude, Michael J. Stechman, Susan Moug, Kathryn McCarthy, Jonathan Hewitt, Ben Carter, Phyo Kyaw Myint

**Affiliations:** 1Aberdeen Royal Infirmary, Aberdeen AB25 2ZN, Scotland, UK; eilidh.bruce@nhs.net; 2Institute of Applied Health Science, University of Aberdeen, Aberdeen AB25 2ZN, Scotland, UK; 3Royal Alexandra Hospital, Paisley PA2 9PJ, Scotland, UK; Fenella.barlow-pay@nhs.net (F.B.-P.); susanmoug@nhs.net (S.M.); 4Department of Biostatistics & Health Informatics, King’s College London, London WC2R 2LS, UK; roxanna.short@kcl.ac.uk (R.S.); ben.carter@kcl.ac.uk (B.C.); 5Salford Royal NHS Trust, Salford M6 8HD, UK; Arturo.Vilches-Moraga@srft.nhs.uk (A.V.-M.); angeline.price@srft.nhs.uk (A.P.); 6Glasgow Royal Infirmary, Glasgow G4 0SF, Scotland, UK; aine.mcgovern@nhs.net; 7North Bristol NHS Trust, Bristol BS10 5NB, UK; philip.braude@nbt.nhs.uk (P.B.); kathryn.mccarthy@nbt.nhs.uk (K.M.); 8School of Medicine, Cardiff University, Cardiff CF10 3AT, Wales, UK; Michael.stechman@wales.nhs.uk (M.J.S.); Hewittj2@cardiff.ac.uk (J.H.)

**Keywords:** covid-19, SARS-CoV-2, non-steroidal anti-inflammatory drugs, NSAIDs

## Abstract

Coronavirus disease 2019 (COVID-19) infection causes acute lung injury, resulting from aggressive inflammation initiated by viral replication. There has been much speculation about the potential role of non-steroidal inflammatory drugs (NSAIDs), which increase the expression of angiotensin-converting enzyme 2 (ACE2), a binding target for severe acute respiratory syndrome coronavirus 2 (SARS-CoV-2) to enter the host cell, which could lead to poorer outcomes in COVID-19 disease. The aim of this study was to examine the association between routine use of NSAIDs and outcomes in hospitalised patients with COVID-19. This was a multicentre, observational study, with data collected from adult patients with COVID-19 admitted to eight UK hospitals. Of 1222 patients eligible to be included, 54 (4.4%) were routinely prescribed NSAIDs prior to admission. Univariate results suggested a modest protective effect from the use of NSAIDs, but in the multivariable analysis, there was no association between prior NSAID use and time to mortality (adjusted HR (aHR) = 0.89, 95% CI 0.52–1.53, *p* = 0.67) or length of stay (aHR 0.89, 95% CI 0.59–1.35, *p* = 0.58). This study found no evidence that routine NSAID use was associated with higher COVID-19 mortality in hospitalised patients; therefore, patients should be advised to continue taking these medications until further evidence emerges. Our findings suggest that NSAID use might confer a modest benefit with regard to survival. However, as this finding was underpowered, further research is required.

## 1. Introduction

The pattern of acute lung injury seen in severe acute respiratory syndrome coronavirus 2 (SARS-CoV-2) infection—commonly referred to as coronavirus disease 2019 (COVID-19)—is thought to be a result of aggressive inflammation initiated by viral replication; however, the exact pathophysiology behind this phenomenon remains largely unknown [1]. Following recognition that the angiotensin-converting enzyme 2 receptor (ACE2) serves as a binding site for SARS CoV-2 to enter the host cell, a number of European authorities, including those in France and Belgium, issued federal reports suggesting that the use of non-steroidal inflammatory drugs (NSAIDs) in the presence of COVID-19 might adversely affect patients’ clinical course and recovery [2]. There remains, however, a considerable uncertainty regarding the use of common NSAIDs and their effect on COVID-19. 

NSAIDs are one of the most commonly prescribed and used pain medications worldwide, for both acute pain and chronic conditions such as rheumatological diseases and osteoarthritis [3,4], with analgesic, anti- inflammatory and anti-pyretic properties. The risks of NSAID use have been well studied in the general population, with particular focus on the association between their long-term use and increased risk of upper gastrointestinal effects (e.g., ulceration, bleeding), renal impairment and arterial thrombotic events (e.g., myocardial infarction, stroke) [4]. It has been speculated that ibuprofen may upregulate the cellular expression of ACE2 [5], and in the context of COVID-19, it is therefore postulated that NSAID use could result in a higher viral infective load in the respiratory tract. More recent work has also associated NSAIDs with high complication rates (effusion, empyema, dissemination of infection) after acute respiratory tract infection [6,7], via NSAID-mediated cyclo-oxygenase (COX) inhibition. It has been proposed that the inhibition of COX enzymes reduces the recruitment of polymorphonuclear cells and inhibits the synthesis of lipoxins and resolvins, ultimately delaying the resolution of inflammation. 

However, there exists a conflict of opinion within the literature with regard to coronaviruses, and very little research has been conducted to date. In vitro studies of the earlier SARS-CoV infection in animal models and human lung epithelium found indomethacin to have potent antiviral activity by inhibiting viral RNA synthesis [8], an effect that was independent of COX inhibition. Whilst the anti-inflammatory properties of NSAIDs are well known, it is unclear what, if any, their effect is on outcomes in patients with acute respiratory tract infection. 

Available published evidence describing outcomes for patients with COVID-19 infection who are NSAID users is specifically lacking. The primary aim of this study is therefore to examine the association between prior routine use of NSAIDs and mortality and length of stay in patients with COVID-19 admitted to hospital. 

## 2. Methods 

### 2.1. Study Population 

The study population was drawn from the COPE (COVID-19 in Older People) study [9]. The COPE study is a multicentre, observational study governed by the Older Persons Surgical Outcome Collaborative (OPSOC; www.opsoc.eu), our existing academic network of clinical centres. OPSOC runs a well-established programme of research with experience in collecting epidemiological data for both academic and service evaluation purposes. In the current study, a total of eight UK centres were included. They were all involved in the delivery of unscheduled, in-patient treatment to patients with COVID-19. Data were gathered between 6 March and 28 April 2020. Patient outcomes were up to 28 April 2020. A detailed description of the study protocol and data collection methods has been previously reported [9].

In brief, data collection was undertaken prospectively using a standardised, computerised case report. This was populated after reviewing patients’ individual paper records, prescription administration records and information from electronic records. Each site’s principal investigator supervised the study personnel collecting data at a local level, all of whom had completed data collection training prior to their involvement in the study. Data were recorded securely at each site by adherence to data protection policy, and collated data were ultimately transferred in an anonymised format to King’s College London for statistical analysis. 

### 2.2. Participants

Patients aged 18 years or older who were admitted to hospital with a clinical or laboratory confirmed diagnosis of COVID-19 were included. Clinical diagnoses were made by clinicians at each site, based on signs, symptoms and/or radiological appearance consistent with COVID-19, whilst laboratory-confirmed diagnoses required positive PCR results from a swab test for SARS-CoV-2. There were no exclusion criteria. 

### 2.3. Endpoints

The primary endpoint was the time to mortality from the date of admission or date of diagnosis, when the patient was diagnosed with COVID-19 five or more days after admission. Secondary outcomes included day 7 mortality and the time from admission or diagnosis (if diagnosed a minimum of five days after admission) to discharge (length of stay).

### 2.4. Exposure

Data on the use of NSAIDs, including the number, type and dose of anti-inflammatory that each patient was taking prior to admission, were collected from admission records and from online GP prescription records. NSAIDs included in the data collection were: propionic acid derivatives such as ibuprofen and naproxen, diclofenac, an acetic acid derivative and selective COX-2 inhibitors such as celecoxib. Topical NSAIDs such as ibuprofen gels were not included due to their low level of systemic absorption and consequential limited systemic effects [10,11]. Low-dose aspirin was not included as an NSAID in our data collection, as although it is a COX inhibitor, its effects are primarily anti-platelet at low doses, with minimal anti-inflammatory effects. History of coronary artery disease was included in the covariate data collection and adjusted for; the majority of these patients were taking aspirin. 

### 2.5. Covariates 

Additional clinical demographics collected included: age, sex, smoking status (current, previous, never), C-reactive protein (CRP) levels on admission, reduced renal function (estimated glomerular filtration rate (eGFR) < 60 on admission) and the presence of comorbidities including diabetes mellitus, hypertension, and coronary artery disease (CAD). 

### 2.6. Statistical Analysis 

Baseline demographic and clinical characteristics were compared by in-hospital mortality status and NSAID consumption versus lack of it. Time-to-event outcomes (death or discharge) were analysed with mixed-effects multivariable Cox’s proportional baseline hazards models. The analyses were fitted with a random intercept to account for hospital variation and adjusted for the base model of: NSAID prescribed (yes/no), patient age group, sex, smoking status, CRP levels, diabetes, hypertension, coronary artery disease, reduced renal function (eGFR < 60). The adjusted hazard ratios (aHR) were estimated with associated 95% confidence intervals (95% CI). The baseline proportionality assumption was tested visually with log-log residuals. Each time-to-event analysis was reported with a Kaplan–Meier survival plot. Day 7 mortality was analysed using a mixed-effects multivariable logistic model, fitting each hospital as a random intercept effect, and adjusted with covariates consistent with the time-to-event analyses. The adjusted odds ratios (aOR) were estimated and presented with corresponding 95% confidence interval (95% CI). Missing data were explored for patterns of missingness. Subgroup analyses were carried out to explore potentially moderating effects of NSAID use within different subgroups stratified by age group, sex, smoking status, diabetes, hypertension, coronary artery disease and renal impairment. Analysis was carried out using Stata version 15 [12]; Kaplan–Meier survival plots were visualised in R [13], with packages survival [14] and survminer [15]. 

Permission to undertake the study was received from The Health Research Authority (20/HRA/1898); Cardiff University acted as study sponsor. 

## 3. Results

Data were collected from 1222 patients with COVID-19 across eight UK sites, of whom 56.5% (*n* = 690) were male. Of these patients, 4.4% (*n* = 54) were prescribed routine NSAIDs prior to admission. There were 19 patients with missing CRP data, inputted as CRP < 40, and a further 20 patients with missing smoking status, who were recorded as ‘never smokers’. Overall in-hospital mortality was 29.3% (*n* = 358), varying from 12.2 to 43.9% across hospital sites. In-hospital mortality was 25.9% (*n* = 14) for the NSAID users and 29.5% (*n* = 344) among the non-users (*p* = 0.578). In-hospital mortality was higher in older age groups (39% in patients aged ≥80 years; 34.3% in those aged 65–79 years; 12.9% in patients aged <65 years) and for patients with co-morbidities including diabetes (32.4% in patients with diabetes vs. 28.1% in those without), hypertension (32.6% vs. 25.7%), coronary artery disease (39.2% vs. 26.3%) and reduced renal function (eGFR < 60) on admission (38.7% vs. 23.1%). A complete breakdown of demographics and clinical characteristics by in-hospital mortality is shown in Table 1. 

Overall, NSAID use in the study population was 4.4% (*n* = 54), ranging from 2.6 to 18.6% across the eight hospital sites. Pre-admission NSAID use was higher in the younger age groups (<65 years, 6.5%; 65–79 years, 4.2%; ≥ 80 years, 2.8%) and lower in patients with co-morbidities including diabetes (3.3% in diabetic patients vs. 4.9% in non-diabetics), hypertension (2.8% vs. 6.1%), coronary artery disease (3.3% vs. 4.8%) and reduced renal function (3.0% vs. 5.4%). The routine prescription of NSAIDs was higher for patients with an elevated CRP level on admission; in fact, 5.1% of patients with an elevated CRP level (>40 mg dL^−1^) were prescribed routine NSAIDs prior to admission, compared to 3% of those without an elevated CRP value. Full patient demographics and clinical characteristics by NSAID use are shown in Table A1.

### Outcome Analysis

The primary endpoint was time to mortality. The Kaplan–Meier Survival plot suggested a modest protective effect of NSAID use (Figure 1, Kaplan–Meier curves), but in the crude analysis with 95% CI, we found no association between the routine use of NSAIDs and time to mortality, hazard ratio (HR) = 0.82 (95% CI 0.48–1.40, *p* = 0.46, Table 2; indicated by overlapping shaded areas in Figure 1). Important covariates which have previously been linked to poorer outcomes in COVID-19 patients were associated with a reduced time to mortality. These included: advancing age (compared to patients aged <65 years: patients aged 65–79 years, HR = 3.21, 95% CI 2.29–4.51; patients aged over 80 years, HR = 3.94, 95% CI 2.52–5.50), reduced renal function (eGFR < 60) on admission (HR = 1.80, 95% CI 1.45–2.24, *p* < 0.001) and the presence or history of coronary artery disease (HR = 1.47, *p* = 0.001) and hypertension (HR = 1.27, 95% CI 1.03–1.58, *p* = 0.03). 

In the multivariable analysis, there was no association between pre-admission NSAID use and time to mortality (adjusted HR (aHR) = 0.89, 95% CI 0.52–1.53, *p* = 0.67, Table 2). Advancing age (65–79 years vs. under 65 years, aHR = 3.14, 95% CI 2.20–4.48, *p* < 0.001; over 80 vs. under 65; aHR = 4.00, 95% CI 2.81–5.71, <0.001), elevated CRP level (>40 mg/dL; aHR = 2.75, 95% CI 2.01–3.76, *p* < 0.001) and reduced renal function on admission (eGFR < 60; aHR = 1.40, 95% CI 1.11–1.75, *p* = 0.004) were associated with mortality.

In the analysis for secondary outcomes, no association was found between pre-admission NSAID use and day-7 mortality (aOR = 0.79, *p* = 0.60, Table 3) or time to discharge (aHR = 0.89, 95% CI 0.59–1.35, *p* = 0.58, Table 3). Day-7 mortality was associated with older age (compared to patients aged <65 years: 65–79 years aOR = 3.80, *p* < 0.001; ≥80 years aOR = 5.14, *p* < 0.001), elevated CRP level on admission (aOR = 4.91, *p* < 0.001) and reduced renal function (aOR = 2.02, *p* < 0.001). An increased length of stay was associated with older age and elevated CRP on admission (see Table 3).

Figure A1, Figure A2 and Figure A3 show subgroup analysis for patients prescribed NSAIDs in relation to the three endpoints, i.e., time to mortality, day-7 mortality and length of stay. Hazards ratios and odds ratios were adjusted for age group, sex, smoking status and co-morbidities. Due to the small number of NSAID users in our study population, these analyses were underpowered and should be interpreted with caution and with a large degree of uncertainty. 

## 4. Discussion

To the best of our knowledge, this study is the first to report on routine NSAID use and outcomes in hospitalized patients with COVID-19. We found that the routine use of NSAIDs might confer a modest survival benefit and is not associated with poorer outcomes. Uncertainty and speculation have surrounded this topic, and existing literature primarily references experience of these drugs in the setting of previous respiratory virus outbreaks [16]. Therefore, our findings provide novel information at a time where there is a significant lack of evidence and high demand for knowledge. 

Whilst systematic reviews have concluded that there is insufficient evidence to determine the effect of NSAIDs in COVID-19 patients [16,17,18,19], advising against drastic changes to drug regimens, alternative literature recommends that NSAIDs should be avoided until evidence emerges [20]. Many of the studies referenced within literature reviews are in relation to respiratory viruses other than SARS-CoV-2. It is thought that the pathophysiology and transmission of COVID-19 shows differences in behaviour even from other viruses within its family, such as SARS-CoV and Middle Eastern respiratory syndrome (MERS) virus [19]. Therefore, the findings of these studies may not be applicable to COVID-19. However, very recently Rinott et al. [21] published a retrospective cohort study of 403 cases of COVID-19 patients, recruiting those who used “any medication containing ibuprofen, paracetamol or dipyrone starting a week before diagnosis of COVID-19”. Their conclusions therefore address the issue of acute use of NSAIDs and found no significant association between ibuprofen usage and clinical outcomes (including mortality) when comparing the NSAID group with cohorts using paracetamol or no antipyretic.

Pre-admission NSAID use was lower in older age groups, which is in line with a recent European study of NSAID epidemiology [22]. NSAIDS account for 25% of adverse drug events reported in the United Kingdom [23], and it is thought that the elderly are more susceptible to the adverse effects of these drugs [4]. Conversely, the higher percentage of NSAID use in younger adults in this study may reflect their relative co-morbidity and subsequent susceptibility to hospitalisation as a result of COVID-19.

Pre-admission NSAID use was also found to be lower in patients with co-morbidities across all age groups. This may be due to the relative contraindication of NSAIDs in these patients [24] and potential for drug interactions [25]. This low prevalence of NSAIDs routine use may be related to the presence of background respiratory illness in a high proportion of patients with COVID-19, which was a contraindication to their use. In the analysis of biochemical markers, the routine prescription of NSAIDs was associated with elevated CRP level on admission. This was expected and is likely a reflection of pre-existing inflammatory conditions, in addition to COVID-19 infection, that required regular NSAID usage in this patient group.

The limitations of this study include a relatively small number of NSIAD users in our cohort; thus, this study may be underpowered. It is possible that patients who had been using over-the-counter NSAIDs were not captured. Conversely, as we collected data from prescriptions, it is possible that some of the patients prescribed NSAIDs were not actually taking them in the days or weeks leading up to admission. We did not collect data on indication for or duration of use of the agents, hence we cannot rule out possible confounding by indication. 

There are intrinsic limitations associated with any observational study. However, the prospective relationship described reduces the possibility of reverse causality, whilst unselected cohort design reduced the selection bias. This study did not take into consideration a proportion of patients who were discharged from or died in the Emergency Department or those patients with COVID-19 who remained in the community and were not treated in hospital. It is worth highlighting that our results specifically apply to the routine use of NSAIDs by patients who then develop COVID-19. Our results therefore do not signify the safety of an acute use of NSAIDs in the context of COVID-19. This calls for future work examining the relationship between NSAID usage, by indication and duration as well as routine or acute usage, and mortality of COVID-19 patients in a high-powered study. 

Our study has several strengths. It was a multi-centre, prospective study, providing a large study population, which included patients from a range of specialties including medicine, surgery and medicine for the elderly. Through the inclusion of eight UK sites, the data are representative of populations across England, Scotland and Wales. Data were collected by trained personnel from paper and electronic records and prescription charts, which resulted in minimal missing data. 

This study has a number of direct implications on both clinical practice and research. Based on our results, patients and clinicians should not associate the routine use of NSAIDs with an increased risk of mortality in COVID-19 patients; therefore, we recommend that patients continue to comply with their baseline drug regimen. The association between NSAIDs and mortality in patients with COVID-19 warrants investigation via randomised controlled trials, as a method of further examining the potential beneficial effects that this study has suggested. Indeed, the LIBERATE Trial in COVID-19 [26] is an ongoing, multi-centre randomised controlled trial for lipid ibuprofen treatment versus standard of care for acute hypoxic respiratory failure due to COVID-19. This is primarily in a critical care setting. However, future investigation should not only concentrate on the use of NSAIDs as a therapeutic option but also continue to explore the effects of pre-admission NSAID use on outcomes and mortality in the general population affected by COVID-19. Future, more highly powered studies would more reliably comment on the association of type, dose and duration of NSAID therapy with these important outcomes.

Our findings show no significant negative effect of routine NSAID use on mortality in patients with COVID-19. Indeed, a modest beneficial effect of routine NSAID use on mortality may well exist, though it cannot be concluded from the evidence presented here. This study provides novel information about the impact of NSAID use on outcomes of COVID-19, during a pandemic characterised by much uncertainty. Further evidence is required to explore this possible correlation and subsequently guide public health policy.

## Figures and Tables

**Figure 1 jcm-09-02586-f001:**
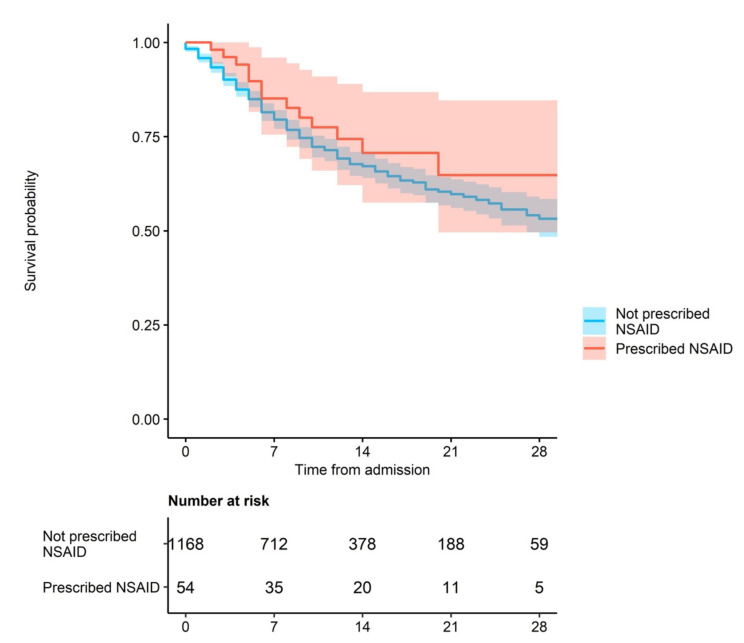
NSAID use and mortality (shaded areas indicate 95% confidence intervals).

**Table 1 jcm-09-02586-t001:** Demographics, comorbidities and non-steroidal inflammatory drugs (NSAID) usage by in hospital mortality.

	Alive	Dead	Total	*p*-Value ^&^
	(*n* = 864)	(*n* = 358)	(*n* = 1222)	
**NSAID Prescription**				0.578
No	824 (70.6)	344 (29.5)	1168 (95.6)	
Yes	40 (74.1)	14 (25.9)	54 (4.4)	
**Sites**				<0.001
Hospital A	119 (77.8)	34 (22.2)	153 (12.5)	
Hospital B	33 (76.7)	10 (23.3)	43 (3.5)	
Hospital C	108 (87.8)	15 (12.2)	123 (10.1)	
Hospital D	254 (66.8)	126 (33.2)	380 (31.1)	
Hospital E	76 (67.9)	36 (32.1)	112 (9.2)	
Hospital F	138 (56.1)	108 (43.9)	246 (20.1)	
Hospital G	100 (87.0)	15 (13.0)	115 (9.4)	
Hospital H	36 (72.0)	14 (28.0)	50 (4.1)	
**Age**				<0.001
Under 65 years	337 (87.1)	50 (12.9)	387 (31.7)	
65 to 79 years	266 (65.7)	139 (34.3)	405 (33.1)	
Over 80 years	261 (60.7)	169 (39.3)	430 (35.2)	
**Sex**				0.625
Female	380 (71.4)	152 (28.6)	532 (43.5)	
Male	484 (70.1)	206 (29.9)	690 (56.5)	
**Smoking Status**				0.049
Never smokers	453 (73.2)	166 (26.8)	619 (50.7)	
Ex-smokers	325 (66.7)	162 (33.3)	487 (39.9)	
Current smokers	74 (77.1)	22 (22.9)	96 (7.8)	
Missing	12	8	20	
**Diabetes**				0.100
No	638 (71.9)	249 (28.1)	887 (72.6)	
Yes	223 (67.6)	107 (32.4)	330 (27.0)	
Missing	3	2	5	
**Hypertension**				0.008
No	448 (74.3)	155 (25.7)	603 (49.4)	
Yes	414 (67.4)	200 (32.6)	614 (50.3)	
Missing	2	3	5	
**Coronary Artery disease**				<0.001
No	696 (73.7)	248 (26.3)	944 (77.3)	
Yes	166 (60.8)	107 (39.2)	273 (22.3)	
Missing	2	3	5	
**Elevated CRP (>40)**				<0.001
No	282 (84.4)	52 (15.6)	334 (27.3)	
Yes	571 (65.7)	298 (34.3)	869 (72.7)	
Missing	11	8	19	
**Renal function (eGFR < 60)**				<0.001
No	568 (76.9)	171 (23.1)	739 (60.5)	
Yes	290 (61.3)	183 (38.7)	473 (38.7)	
Missing	6	4	10	

^&^ Chi-square test was carried out between clinical characteristics and mortality.

**Table 2 jcm-09-02586-t002:** Time to mortality. Crude and multivariable analysis.

	Crude Hazard Ratio (HR)		Adjusted Hazard Ratio (aHR) ^&^	
	(*n* = 1181)		(*n* = 1167)	
	HR, (95% CI)	*p*-Value	aHR, (95% CI)	*p*-Value
**NSAID**	0.82 (0.48–1.40)	0.46	0.89 (0.52–1.53)	0.67
**Age**				
Under 65	-Ref-		-Ref-	
65 to 79	3.21 (2.29–4.51)	<0.001	3.14 (2.20–4.48)	<0.001
Over 80	3.94 (2.82–5.50)	<0.001	4.00 (2.81–5.71)	<0.001
**Sex**				
Female	-Ref-		-Ref-	
Male	0.88 (0.71–1.10)	0.25	0.88 (0.70–1.11)	0.28
**Smoking status**				
Never				
Ex-smokers	1.24 (1.0–1.55)	0.06	1.02 (0.80–1.28)	0.92
Current smokers	0.90 (0.56–1.42)	0.62	1.11 (0.68–1.82)	0.66
**Elevated CRP (>40)**	2.24 (1.65–3.05)	<0.001	2.75 (2.01–3.76)	<0.001
**Patients with diabetes**	1.09 (0.87–1.38)	0.45	1.03 (0.81–1.32)	0.80
**Patients with CAD**	1.47 (1.16–1.87)	0.001	1.09 (0.84–1.40)	0.53
**Patients with hypertension**	1.27 (1.03–1.58)	0.03	0.97 (0.77–1.22)	0.81
**Patients with reduced renal function (eGFR < 60)**	1.80 (1.45–2.24)	<0.001	1.40 (1.11–1.76)	0.004

^&^ The multivariable mixed-effects analysis was adjusted for age group, sex, smoking, C--reactive protein (CRP) level, diabetes, coronary artery disease (CAD), hypertension and renal function; eGFR, estimated glomerular filtration rate.

**Table 3 jcm-09-02586-t003:** Day-7 mortality and time to discharge. Multivariable analysis.

	Adjusted Odds Ratio (OR)		Adjusted Hazard Ratio (aHR) ^&^	
	(*n* = 1158)		(*n* = 1167)	
	aOR, (95% CI)	*p*-Value	aHR, (95% CI)	*p*-Value
**NSAID**	0.79 (0.32–1.92)	0.602	0.89 (0.59–1.35)	0.58
**Age**				
Under 65	-Ref-		-Ref-	
65 to 79	3.80 (2.26–6.37)	<0.001	0.76 (0.61–0.95)	0.02
Over 80	5.14 (3.04–8.69)	<0.001	0.56 (0.44–0.73)	<0.001
**Sex**				
Female	-Ref-		-Ref-	
Male	0.81 (0.57–1.14)	0.227	0.92 (0.76–1.11)	0.37
**Smoking status**				
Never				
Ex-smokers	1.17 (0.83–1.67)	0.372	0.95 (0.78–1.15)	0.61
Current smokers	1.12 (0.54–2.33)	0.765	1.04 (0.73–1.48)	0.84
**Elevated CRP (>40)**	4.91 (2.99–8.06)	<0.001	0.69 (0.57–0.84)	<0.001
**Patients with diabetes**	1.04 (0.71–1.52)	0.838	0.84 (0.68–1.04)	0.12
**Patients with CAD**	1.46 (1.00–2.13)	0.051	1.12 (0.88–1.42)	0.36
**Patients with hypertension**	0.78 (0.55–1.10)	0.157	0.95 (0.79–1.16)	0.63
**Patients with reduced renal function (eGFR < 60)**	2.02 (1.42–2.86)	<0.001	0.92 (0.75–1.13)	0.58

^&^ The multivariable mixed-effects analysis was adjusted for age group, sex, smoking, CRP level, diabetes, CAD, hypertension and renal function.

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
