# Peer review of "Prior Routine Use of Non-Steroidal Anti-Inflammatory Drugs (NSAIDs) and Important Outcomes in Hospitalised Patients with COVID-19"

_jcm, 2020, doi:10.3390/jcm9082586_

Round 1

Reviewer 1 Report

The aim of this study was to examine the association between routine use of NSAIDs and outcomes in hospitalized patients with COVID-19 infection. The topic could be of interest, but these results are unlikely to change the current clinical practice because of some major issues with methodology.

Major concerns:

  1. The authors refer to ‘routine use of NSAIDs’ as the exposure of interest. However, they do not define the clinical indication for the NSAIDs use. This could result in confounding by indication, especially with underpowered studies. Moreover, the duration of NSAIDs use is not provided either, hence one cannot distinguish between sporadic use as a painkiller rather than disease-modifying drug for osteoarthritis or other rheumatic and musculoskeletal diseases.
  2. Only 4.4% of patients (n=54) were prescribed routine NSAIDs prior to admission. I think this number is too low to detect meaningful and significant associations with the outcomes of interest. 
  3. Both patients with clinical and laboratory-confirmed diagnosis of COVID-19 were included. I was unable to find the study protocol that is referenced (ref #9), thus data collection methods are unknown. However, from the public infopage (https://www.acpgbi.org.uk/news/cope-study-covid-19-in-older-people/) looks like the researchers originally considered only lab confirmed COVID-19 diagnosis. Is this a post-hoc analysis of the COPE study? How many patients belonged to each group ? What were their clinical characteristics? In this study one cannot be sure that the participants had COVID-19 pneumonia, flu, COPD, heart failure or other several COVID-19 mimickers.

Other remarks:

Introduction

  • The possible effect of NSAIDs on COVID-19 pathology (ACE2 receptor upregulation, COX inhibition, direct antiviral activity) are split between the beginning and the end of the introduction, and it should be compacted in a single paragraph.

Methods

  • Page 4, line 121-122: no data was collected regarding the corticosteroid use. Corticosteroids are often prescribed when there is a relative contraindication to NSAIDs, especially in elderly patients and in those with coronary artery disease, systemic arterial hypertension and decreased renal function. Noteworthy, the corticosteroid dexamethasone has been proven to reduce mortality from COVID-19. 
  • Page 4, line 124: data on aspirin use were not collected, rather it was inferred from the number of patients diagnosed with coronary artery disease. However, aspirin could be prescribed also for primary CVD prevention. There is also a significant interaction between ibuprofen and aspirin, the latter being inhibited by ibuprofen.
  • Page 4, line 129: the decision to set the cut off at 40 mg/L seems rather arbitrary, while it should be an important point in a study dealing with anti-inflammatory drugs.
  • Missing data were explored for patterns of missingness, but which method was used for missing data analysis is not provided.

RESULTS

  • Overall in-hospital mortality was very different across hospital sites, suggesting differences in patients’ characteristics and COVID-19 management which could affect the outcomes (mortality). 

DISCUSSION: 

  • The discussion tries to cover too many aspects of the topic and should be shortened by 30-50%.
  • Page 11, lines 249-253: speculative (no supporting data) interpretation.
  • Page 11, line 262: I suggest to underline here that no data were collected on inflammatory rheumatic conditions which prompt NSAIDs usage, as written on page 2, line 62. Moreover, there is growing evidence - it should be referenced - that inflammatory rheumatic diseases confer a  lower risk of cytokine storm from COVID-19 than the general population, but it unclear whether this is due to the intrinsic immune dysregulation or the effect of anti-inflammatory (NSAIDs, glucocorticoids) or immune-modulating treatments.
  • Page 11, line 266: it is not clear what ‘background inflammatory disease’ means in this context. Pre-existing inflammatory condition? More severe respiratory COVID-19 related complications? (but there is no data regarding the severity at admission, only mortality and hospitalization duration).

Author Response

thank you for your comments. We attach the responses in word document.

Reviewer 2 Report

I applaud the authors for their efforts to address this timely subject.  It is sorely needed and I suspect will be well received.

Please consider the following:

1) Please cite additional evidence to support the assertion that NSAIDs increase ACE2 levels (lines 56-57). The listed reference for this, Gracia-Ramos AE. Is the ACE2 Overexpression a Risk Factor for COVID-19 infection?, cites the following article as support that NSAIDs increase ACE2 levels: Fang L., Karakiulakis G., Roth M. Are patients with hypertension and diabetes mellitus at increased risk for COVID-19 infection? Lancet Respir Med. 2020;8:e21. However, in this root citation (a correspondence piece), Fang et al does not actually cite evidence that NSAIDs can increase ACE2 levels--and I am not aware of any research that demonstrates this. If a citation can be found to substantiate this claim, that would be great. If not, this portion of your paper should be amended accordingly.

2) When talking about the risks of NSAID use in the introduction, was there a reason to not also list renal side effects (lines 64-65)?

Author Response

thank you for your comments. pls refer to the attached word document in response to your review.

Reviewer 3 Report

Thank you for this work - it's relevant topic and a very well-written piece.

The main point here is as you point out in the limitation, the number of NSAIDs users are very small to draw firm conclusion out of this analysis. Especially, when it is shown only 2.8% of the oldest group were the NSAID users. So, it is good not to indicate any form of association from this small, non-randomised study. 

On Table 1, it maybe a good idea to put additional column to show p-value for the differences between the groups.

On page 10, line 233, please take out 'views'

Author Response

thank you for your comments. Pls refer to the word document attached. 

Reviewer 4 Report

These are my comments about the manuscript " Prior Routine use of Non-Steroidal Anti-Inflammatory Drugs (NSAIDs) and Important Outcomes in Hospitalised Patients with COVID-19" that I have been asked to evaluate.

The topic is of great interest to the medical community during these times of COVID-19, dealing with an important, as yet unresolved issue that has clinical consequences.

Overall, the manuscript is well written.

My main concern is the fact that the authors, as they acknowledge in the limitations, use prescriptions and admission information to evaluate for NSAID use, meaning they will only get at the chronic use. This medication has been speculated by some to aggravate symptoms in COVID-19 patients even when taken as over the counter medication for acute use. This, obviously, is important in a disease manifesting with fever and pain, where many patients will take NSAIDs as an antipyretic, a fact this study does not address. While the fact that chronic use was not associated with worse outcomes might be considered by some as proof that short-term use is safe, I would argue that it does not necessarily signify the safety of acute use. For example, with some medications you can expect tolerance in "off target" effects like increase in ACE 2 receptors, so that paradoxically short term use can be associated with greater effects.

This issue in itself does not mean the manuscript isn't interesting, I merely suggest emphasizing the exact study question answered, in the discussion.

Comments:

Introduction, second paragraph, would add the use of NSAIDs as antipyretics, not only as anti-inflammatories

Methods:

Was any information about short-term use of NSAIDs collected? What about use of NSAIDs during the admission?

Results:

The authors differentiate between clinical and laboratory confirmed COVID-19. Please add this distinction to the results stratified by NSAID use, as these patients may represent different groups.

Given the small number of NSAID users this might well be futile, but was there any association between the dose/type/duration of treatment with NSAIDs and the outcome?

As an intermediate outcome and a surrogate for severity of disease, would it be possible to add the association between NSAID use and need for oxygen in the hospital?

Discussion:

…"This study is the first to report on NSAID use and outcomes…" Would add a recent paper that evaluated any use of ibuprofen and the results of COVID-19 "Rinott et al, " Ibuprofen Use and Clinical Outcomes in COVID-19 Patients" Clin Microbiol Infect. 2020 Jun.

Again, please emphasize the distinction made in the paper, and the fact that it deals with routine use whereas most "press releases" and discussions have surrounded advice to refrain from any use, including acute.

Thank you for allowing me to evaluate your work

Author Response

thank you for your comments. pls refer to the word document attached. 
